# Physiological Predictors of Peak Velocity in the VAM-EVAL Incremental Test and the Role of Kinematic Variables in Running Economy in Triathletes

**DOI:** 10.3390/sports13090316

**Published:** 2025-09-10

**Authors:** Jordi Montraveta, Ignacio Fernández-Jarillo, Xavier Iglesias, Andri Feldmann, Diego Chaverri

**Affiliations:** 1INEFC Barcelona Sports Sciences Research Group (GRCEIB), National Institute of Physical Education of Catalonia (INEFC), University of Barcelona (UB), 08038 Barcelona, Spaindchaverri@gencat.cat (D.C.); 2Institute of Sport Science, University of Bern, 3012 Bern, Switzerland

**Keywords:** VO_2_max, SmO_2_min, NIRS, Vpeak, incremental exercise testing

## Abstract

This study examined the influence of physiological parameters on peak velocity (Vpeak) and of kinematic variables on running economy (RE) during an outdoor incremental VAM-EVAL test completed by eleven national-level triathletes. Maximal oxygen uptake (VO_2_max), ventilatory thresholds, RE, and minimum muscle oxygen saturation (SmO_2_min) were obtained with a portable gas analyzer and near-infrared spectroscopy (NIRS), while cadence, stride length, vertical oscillation, and contact time were recorded with a foot-mounted inertial sensor. Multiple linear regression showed that VO_2_max and SmO_2_min together accounted for 86% of the variance in Vpeak (VO_2_max: r = 0.76; SmO_2_min: r = −0.68), whereas RE at 16 km·h^−1^ displayed only a moderate association (r = 0.54). Links between RE and kinematic metrics were negligible to weak (r ≤ 0.38). These findings confirm VO_2_max as the primary determinant of Vpeak and suggest that SmO_2_min can be used as a complementary, non-invasive marker of endurance capacity in triathletes, measurable in the field with portable NIRS. Additionally, inter-individual differences in cadence, stride length, vertical oscillation, and contact time suggest that kinematic adjustments are not universally effective but rather highly individualized, with their impact on RE likely depending on each athlete’s specific characteristics.

## 1. Introduction

Running performance is influenced by physiological [1], biomechanical [2], and psychological factors [3]. From a physiological perspective, the determinants of performance are maximal oxygen consumption (VO_2_max), running economy (RE), and physiological thresholds [1]; this triad of performance determinants is related to the oxidative capacity of muscle tissue [4]. VO_2_max marks the upper limit of ATP resynthesis through oxidative phosphorylation metabolism [5], RE is defined as the volume of oxygen an athlete needs to cover a distance at a given speed [1], and metabolic thresholds influence the duration for which a % VO_2_max can be sustained [6]. The oxidative capacity of muscle fibers refers to the mitochondria’s ability to supply ATP using oxygen, which, in turn, provides the necessary energy to sustain activity [7]. VO_2_max is determined by the different components of the oxygen transport system [8,9] and is primarily limited by central factors such as cardiac output, blood volume, and hemoglobin mass [10]. However, it has also been shown to correlate with mitochondrial density [11] and the oxidative capacity of muscle tissue [12]. Physiological thresholds are influenced by the oxidative capacity of muscle tissue [1], associated with the proliferation of certain enzymes at the mitochondrial level [13]. Regarding RE, numerous studies reference the factors that affect this performance determinant, suggesting that RE reflects the interaction of various physiological, biomechanical, anthropometric, and kinematic factors [14,15]. It has been shown that certain physiological aspects impact RE, such as the oxidative capacity [11], capillarization, and myoglobin [7] of skeletal muscle. Another aspect to consider that may affect RE is muscle fiber composition, with suggestions that better RE may be associated with a higher percentage of type 1 fibers [14], as these are more efficient at frequencies between 60 and 120 rpm [1,7]. This range encompasses the cadences used in training and competition in running, despite interindividual differences associated with this kinematic parameter [16].

Muscle oxygen saturation (SmO_2_) provides detailed insights into the balance between oxygen supply and demand by measuring changes in oxygenated and deoxygenated hemoglobin and myoglobin concentrations in muscle tissue [17]. While other metrics, such as deoxygenated hemoglobin and myoglobin (deoxy[heme]), are frequently employed as proxies for oxygen extraction [18], SmO_2_ is comparatively less influenced by blood volume changes [19,20]. It can also be measured non-invasively during sports activity using continuous-wave near-infrared spectroscopy (NIRS) devices [21], reported on a 0–100% scale. Feldmann et al. [22] support the device-level validation of this scale, while more recent studies have demonstrated the application and external validation of SmO_2_ dynamics in field protocols through comparisons with in-dependent physiological markers such as lactate thresholds and EMG activity [23,24]. Nevertheless, NIRS signal can be affected by various factors, such as adipose tissue thickness (ATT) [25], muscle tissue heterogeneity [17], or skin blood flow/volume [26], which tend to elevate SmO_2_ values, since the measurement is influenced by the oxygenation status of hemoglobin in less metabolically active tissues, such as the skin or adipose tissue. Muscle oxidative capacity, which connects the three main performance determinants, has traditionally been limited to invasive or costly assessments such as biopsy or P MRS, but during the last 20 years, advances have been made. First, Motobe et al. [27] developed a non-invasive approach using NIRS to infer muscle oxidative capacity based on the muscle oxygen consumption recovery rate constant (k). This approach was modified almost 10 years later [28], and lately, Pilotto et al. [29] have developed protocols for estimating muscle oxidative capacity and muscle diffusing capacity. The protocols of intermittent occlusions represent a significant advancement compared to the invasive methods implemented some time ago, but they are not efficient options for monitoring the evolution of the oxidative capacity of muscle tissue over time in the context of sports training. In this context, minimum muscle oxygen saturation (SmO_2_min) has been observed to remain consistent within a session upon reaching exhaustion and can predict task failure during high intensity efforts [30]. This marker has been associated with improved endurance [31] and maximal incremental performance [11]. Furthermore, the capacity to deoxygenate correlated significantly with VO_2_peak [32,33], and it appears that the capacity to reach a lower SmO_2_min correlates with performance [34,35,36].

Coaches commonly assess two key running intensities during track tests: peak velocity (Vpeak) and maximal aerobic speed (MAS) [37]. Vpeak represents the highest velocity achieved during a test, whereas MAS refers to the minimal speed required to elicit VO_2_max [38]. These two measures should not be viewed as interchangeable representations of a single concept [39]. Unfortunately, this distinction is often overlooked, and Vpeak is frequently used as a surrogate for MAS [40,41,42]. Both velocities result from the interaction of physiological performance determinants specific to each athlete [1,7]. However, Vpeak is likely to involve a greater contribution from the glycolytic energy system than MAS [39,43]. Despite their physiological differences, both Vpeak and MAS are closely associated with running performance [39,44]. Specifically, regarding triathlon, Vpeak has shown a very strong correlation with overall triathlon performance [45], whereas, to our knowledge, the correlation between triathlon performance and MAS has not been directly assessed. Additionally, Vpeak may be considered a more practical performance marker, as it does not require the simultaneous measurement of oxygen consumption needed to determine MAS.

Establishing a link between the mentioned physiological elements (VO_2_max, VT1, VT2, RE and SmO_2_min) and Vpeak can be beneficial for coaches and athletes to identify the physiological qualities that may influence performance. Additionally, linking the kinematic aspects of running to RE may enhance our understanding of how movement variables influence the oxygen cost of running. Wiecha et al. [46] found that velocity at VT2 and VO_2_max were the strongest predictors of Vpeak. The sample consisted of 4001 recreational endurance athletes, and the derived equations predicted Vpeak accurately in this group. Rather than applying these equations to other populations, such as higher-level athletes or triathletes, we should study those cohorts to determine which variables most strongly predict Vpeak and develop or recalibrate models accordingly. In parallel, there is a need to investigate whether NIRS-derived metrics such as SmO_2_min predict performance and Vpeak in these higher performance cohorts. Evidence on the relationship between spatiotemporal parameters and RE is mixed: Pizzuto et al. [47] found no significant relationships between spatiotemporal parameters and RE in recreational runners, whereas Leite et al. [48] reported that higher cadence and greater vertical oscillation were associated with increased oxygen cost; for each additional 1 step·min^−1^ and 1 mm of vertical oscillation, VO_2_ rose by 0.09 and 0.10 mL·kg^−1^·min^−1^, respectively. Because both studies were conducted on treadmills, the relationship between RE and spatiotemporal variables should also be examined in real-world, overground contexts such as an athletics track. Taken together, these points underscore the need for ecologically valid, tightly controlled studies and for investigations in specific performance cohorts, such as triathletes, to generate findings that are directly actionable for coaches. To address these gaps and extend evidence beyond treadmill-based studies in recreational runners, the primary objective of this study is to evaluate how VO_2_max, ventilatory thresholds, RE, and SmO_2_min influence Vpeak in national-level triathletes. We hypothesize that higher VO_2_max and lower SmO_2_min will be associated with higher Vpeak. The secondary objective is to analyze the impact of cadence, stride length, vertical oscillation, and contact time on RE in this population. We hypothesize that these kinematic variables will not show significant associations with RE.

## 2. Materials and Methods

### 2.1. Study Design

Observational study evaluating the influence of physiological parameters (VO_2_max, ventilatory thresholds, SmO_2_min and RE) on Vpeak in runners. Additionally, the study examines the effect of kinematic parameters (cadence, stride length, vertical oscillation, and contact time) on RE. The physiological parameters were selected due to their established relationship with endurance performance, while the kinematic parameters provide insights into movement efficiency. Vpeak was chosen as the performance metric because it is a critical determinant of athletic performance in endurance sports.

### 2.2. Participants

Eleven national-level triathletes voluntarily participated in the study. The inclusion criteria were as follows: participants had to be active triathletes with a federation license who competed in national triathlon events. The exclusion criteria included any participant with a cardiac condition or those who were currently injured or had suffered an injury in the last two months. Additionally, the exclusion of individuals with an ATT > 7 mm was necessary to minimize interference with NIRS signal quality [25]. ATT was calculated as 0.5 × the mean skinfold thickness. Although the sample size was limited to eleven national-level triathletes, this homogeneous group was chosen to reduce inter-individual variability and focus on high-performance athletes. A priori sample size considerations indicated that detecting a medium effect (f^2^ = 0.15) in a multiple regression with six predictors would require approximately *n* ≈ 99, whereas a large effect (f^2^ = 0.35) would require *n* ≈ 47. Given the restricted availability of national-level triathletes, we recruited the maximum feasible sample (*n* = 11). All participants signed an informed consent form, and the study was approved by the Clinical Research Ethics Committee of the Catalan Sports Administration (026/CEICGC/2023).

### 2.3. Instruments

Muscle oxygenation was monitored using NIRS. The Moxy monitor (Fortiori Design LLC, Fort Collins, CO, USA) was used to measure SmO_2_, allowing an understanding of the relationship between oxygen supply and utilization in the analyzed muscle. The Moxy device uses four wavelengths of near-infrared light (680, 720, 760, and 800 nm), with the sensor equipped with a single LED and two detectors located at distances of 12.5 mm and 25 mm from the light source. The device was placed on the belly of the right vastus lateralis, precisely halfway between the greater trochanter and the lateral epicondyle of the femur [49]. To maintain the sensor’s position relative to the skin, it was secured with waterproof adhesive tape. Hair in the area was shaved, and participants were instructed to avoid applying moisturizers on the testing day. The Moxy sensor was secured to the leg using the Moxy Light Shield, a flexible polyurethane skirt that fits around the sensor. This accessory blocks ambient light, particularly sunlight passing through the tissue, which could interfere with the measurements. The Moxy Light Shield was fixed in place using waterproof adhesive tape to ensure stability during movement and to maintain effective light shielding. The Moxy device was operated in default mode, cycling through four wavelengths 80 times every 2 s and averaging the readings to produce an output rate of 0.5 Hz. SmO_2_ data were averaged at ten second intervals and the SmO_2_min during the incremental running test was identified from a single data point of a ten-second mean value. Participants wore a dead-space mask (Hans Rudolph Inc., Shawnee, KS, USA) equipped with a bidirectional 28 mm digital turbine. Oxygen (O_2_) and carbon dioxide (CO_2_) concentrations were measured using a Galvanic fuel cell O_2_ sensor and digital infrared CO_2_ sensor, respectively, as part of the Cosmed K5 Wearable Metabolic System (Cosmed S.r.l, Albano Laziale, Rome, Italy). The gas analyzer was calibrated according to the COSMED instructions: a room air calibration, a flow meter calibration with a 3 L syringe, a scrubber calibration, a reference gas calibration using a known gas (16% O_2_, 5% CO_2_) and a delay calibration for the breath-by-breath mode. Breath-by-breath data for oxygen consumption (VO_2_) and carbon dioxide production (VCO_2_) were recorded and then averaged. VO_2_ data were measured on a breath-by-breath basis and averaged at ten second intervals. Ventilatory thresholds were assessed by two independent researchers. VT1 was determined using the following criteria: increase in VE/VO_2_ and end tidal partial pressure of O_2_ (PETO_2_) without concomitant increase in VE/VCO_2_. VT2 was determined using the following criteria: increase in VE/VO_2_ and VE/VCO_2_ with a concomitant decrease in end tidal partial pressure of CO_2_ (PETCO_2_) [50]. If the time values identified by the two researchers differed by 40 s or less, their values were averaged. In cases where the difference exceeded 40 s, a third independent researcher evaluated the ventilatory thresholds. The third researcher’s time value was then compared to those of the initial two researchers. If the third researcher’s value was within 40 s of either initial researcher’s value, the two closest time values were averaged to determine the final ventilatory threshold. This method is similar to the one applied by Okawara et al. [51]. VO_2_max was defined as the highest VO_2_ value maintained for ten seconds. RE was calculated at two stages during the incremental test (at the 12 km·h^−1^ stage and at the 16 km·h^−1^ stage) by averaging VO_2_ during the last thirty seconds of the respective stages [52].

Cadence (CAD), vertical oscillation (VO), contact time (CT), and stride length (SL) were recorded using a Stryd inertial sensor (Stryd Inc., Boulder, CO, USA) placed on the right shoe of the athletes. Stryd is a carbon-fiber-reinforced power meter that attaches to the shoe, weighs 9.1 g, and uses a six-axis inertial motion sensor (three-axis gyroscope and three-axis accelerometer). This device has been considered suitable for evaluating kinematic parameters during running [53]. The following kinematic parameters, cadence, vertical oscillation, contact time, and stride length, were recorded throughout the running incremental test and averaged over 10 s intervals. The kinematic parameters were calculated at two stages during the incremental test (at the 12 km·h^−1^ stage and at the 16 km·h^−1^ stage) by averaging data over the last thirty seconds of the respective stages to compare the results with RE data from the same intervals.

### 2.4. Experimental Procedure

Participants were required to attend on one occasion to complete a single experimental session. The session for each one of the participants took place on a 400 m athletics track in Barcelona from February to March with a temperature and humidity of 21.5 ± 5.8 °C and 72.8 ± 16.2%, respectively. In the 24 h prior to the test, participants were asked to refrain from engaging in high-volume and/or high-intensity sessions, whether swimming, cycling, or running. Athletes were advised to consume a carbohydrate-rich diet during the 48 h leading up to the test. Additionally, they were instructed to ensure that their last meal was consumed approximately 3 h before the test. Participants wore a loose-fitting running shirt and non-compressive shorts. The thickness of the skinfolds was measured on the vastus lateralis using a skinfold caliper (Baty Int., Sheffield, South Yorkshire, UK). A heart rate monitor (Polar H10, Polar Electro, Kempele, Finland) was placed on the participants. Finally, participants were fitted with a portable gas analyzer worn on their back, with the mask size selected to fit the individual’s face size. All participants underwent a VAM EVAL test, a commonly utilized method for assessing aerobic capacity in running, derived from the French term “vitesse aérobie maximale” (VAM), which translates to “maximal aerobic speed,” and EVALuation. This test starts at 8 km·h^−1^ for two minutes, and after that, speed increases by 0.5 km·h^−1^ per minute [44]. Cones were placed every 20 m for the participants to regulate their running pace to the audible signal. The researchers, distributed along the track, visually checked that the subjects maintained the imposed pace. The test was terminated when the participants voluntarily stopped due to exhaustion or when they failed to reach the marked cone twice in succession.

### 2.5. Statistical Analysis

The VO_2_, heart rate (HR), SmO_2_, CAD, VO, CT, and SL data were filtered to remove outliers and averaged every 10 s. Data analysis was performed using Microsoft Excel (version 16.81 24011420), Cosmed Omnia (version 2.3), and Moxy Settings App (version 1.5.5). First, the normality of the data was assessed using the Shapiro–Wilk test at a significance level of *p* < 0.05. To determine the association between Vpeak and the following physiological parameters: VO_2_max, VT1, VT2, RE12, RE16, and SmO_2_min, Pearson correlation coefficients (r) were calculated. The procedure was repeated to assess the association between RE and the following kinematic parameters: CAD, VO, CT, and SL. Correlation was classified as negligible (0.00–0.30), weak (0.30–0.50), moderate (0.50–0.70), strong (0.70–0.90), or very strong (0.90–1.00) [54]. A stepwise multiple linear regression method was used to estimate the relative contributions of the independent variables—VO_2_max, VT1, VT2, RE12, RE16, and SmO_2_min—on the dependent variable (Vpeak). Our collinearity diagnostics resulted in variance inflation factors of <2.0 and tolerance levels of >0.10, indicating acceptable levels of multicollinearity among variables [55]. Statistical analysis was performed using Microsoft Excel (version 16.81 24011420) and JASP (version 0.18.3).

## 3. Results

The participants (*n* = 11) presented the following anthropometric, physiological, and kinematic characteristics (Table 1).

The time course of VO_2_ and SmO_2_ during the VAM EVAL test is shown in Figure 1.

The relationships between Vpeak and the analyzed physiological variables and kinematic variables were assessed using Pearson correlation analysis (Table 2).

The stepwise linear regression method showed that VO_2_max and SmO_2_min explain 86% of the variance of the dependent variable Vpeak (Table 3).

## 4. Discussion

The main findings of this study have been that VO_2_max and SmO_2_min explain 86% of the variation in Vpeak. This high degree of predictive accuracy suggests that the predictors (VO_2_max and SmO_2_min) likely capture important elements influencing Vpeak. The positive β coefficient of VO_2_max indicates that better aerobic capacity is associated with higher Vpeak. The β coefficient related to SmO_2_min is, in this case, negative, indicating that when the athlete reaches a lower SmO_2_min, the Vpeak tends to be higher. On the other hand, the kinematic parameters studied (CAD, VO, CT, and SL) showed a negligible to weak correlation with RE.

The relationship between VO_2_max and Vpeak observed in this study aligns with previous findings reporting a high correlation between VO_2_max and running performance. Noakes et al. [56] reported a comparable correlation between VO_2_max and performance (r = 0.81 vs. r = 0.76). Other studies have demonstrated even higher correlations, such as Costill et al. [57] with r = 0.91 and Farrell et al. [58] with r = 0.89. In contrast to the findings of our study, Stratton et al. [59] reported considerably lower correlations between VO_2_max and 5000 m running performance (r = 0.51–0.55).

At the conclusion of an incremental test, when the Vpeak is recorded, O_2_ demand in the musculature is markedly elevated [60]. This results in proportionally high cardiopulmonary workload and O_2_ supply [61]. Given that O_2_ supply reaches its maximum, a lower minimum skeletal muscle oxygen saturation (SmO_2_min) reflects greater oxygen extraction [11,33]. The limitation of SmO_2_ is that it is expressed as a percentage rather than an absolute quantity, and studies have reported significant inter-individual variability in observed values [25]. However, despite these inherent limitations when comparing SmO_2_ values between individuals, our findings align with those of Jacobs et al. [11], who identified SmO_2_min as the third strongest correlate of maximal incremental power output. Additionally, our findings indicate that time to reach SmO_2_min (TTSmO_2_min) is strongly correlated with time to exhaustion (TTE) in the incremental running test (r = 0.89, *p* < 0.001). These results align with those of Feldmann et al. [30], who concluded that SmO_2_min could be used as a predictor of task failure. Similarly, Kirby et al. [62] reported that the percentage rate of change in SmO_2_ (i.e., a faster desaturation rate) was predictive of TTE. This study provides further evidence of the association between SmO_2_min and performance outcomes [11,34,35,36,63], suggesting its potential as a physiological marker to differentiate athletic performance levels. In line with the findings of Feldmann et al. [33], no significant correlation was observed between SmO_2_min in the vastus lateralis and VO_2_max in runners. In contrast, a strong relationship between these variables has been reported in cycling [33]. This difference highlights how NIRS can provide insights into arterio-venous oxygen difference, likely reflecting the greater contribution of the vastus lateralis to systemic oxygen uptake during cycling, in contrast to its more limited involvement during running [4].

SmO_2_ has been used as a proxy for microvascular oxygen pressure (PmvO_2_) [29]. A lower SmO_2_min would indicate a reduced PmvO_2_. According to the Fick law of diffusion, when PmvO_2_ decreases, the gradient between PmvO_2_ and mitochondrial oxygen pressure (PmitO_2_) also diminishes [17]. At that point, any further decrease in SmO_2_ will likely depend on an increased muscle diffusive capacity (DmO_2_), potentially driven by greater capillarization [29]. A potentially enhanced DmO_2_ in these higher performing individuals may support continued O_2_ extraction despite a reduced PmvO_2_–PmitO_2_ gradient. This is supported by the findings from Villanova et al. [64], who found that international and world class swimmers have greater DmO_2_ than recreational to national swimmers. Based on this evidence, SmO_2_min may be associated with DmO_2_ and oxidative capacity, as a lower SmO_2_min could reflect a higher oxidative capacity (i.e., greater oxygen demand from mitochondria) coupled with a higher DmO_2_. Further studies are needed to directly compare oxidative capacity (e.g., k_high_) [29,64] and DmO_2_ (e.g., k_low_ and Δk) with SmO_2_min to determine whether SmO_2_min can serve as a practical surrogate for oxidative capacity and DmO_2_.

Studies have shown that a higher oxygenated hemoglobin and myoglobin (oxy[heme]) level at exhaustion correlates with a greater proportion of oxidative fibers, reflecting the superior vascular conductance and O_2_ delivery capacity that characterize these fibers [65]. At first glance, these results may seem to conflict with the earlier interpretation; however, oxy[heme] should not be used interchangeably with SmO_2_, as SmO_2_ represents the fraction of oxy[heme] relative to total[heme] (oxy[heme] + deoxy[heme]), thereby integrating deoxy[heme] into the denominator of the calculation. Consequently, higher-performing endurance athletes could present both relatively higher oxy[heme] and lower SmO_2_ values due to greater deoxy[heme] and total[heme] amplitudes [66]. Further studies are needed to clarify whether a lower SmO_2_min reflects higher oxidative capacity coupled with enhanced DmO_2_, or instead indicates the recruitment of a greater number of glycolytic fibers with lower O_2_ delivery capacity. Essentially, this would help differentiate whether a lower SmO_2_min is primarily the result of reduced O_2_ delivery, increased O_2_ demand, or a combination of both.

Different confounding factors could affect SmO_2_min measurements, such as adipose tissue thickness [67]. Temperature-induced changes in skin blood flow may also influence SmO_2_ when short source–detector separation distances (20–25 mm) are used [68]. However, when longer source–detector separations are applied (50 mm), skin blood flow does not appear to affect SmO_2_ measurements during exercise [26]. In addition, it has been reported that, in order to detect meaningful changes in SmO_2_ between two measurements using the MOXY monitor, a difference of at least ~9% is required to consider the improvement real with an 84% probability [69].

While VO_2_max is clearly improved by endurance training, particularly through central cardiovascular adaptations, the specific mechanisms appear to depend on training intensity and structure. High-intensity interval training (HIIT) has been shown to enhance VO_2_max by increasing cardiac output, especially stroke volume, when intervals are sufficiently long (≥60 s) and recovery durations allow for venous return and ventricular filling. In addition, when exercise is performed near or above 90% VO_2_max, the combined stimulation of motor unit recruitment, ventilation, and cardiac output provides a strong physiological drive to improve VO_2_max. Peripheral adaptations may also contribute, as some studies report increases in arteriovenous O_2_ difference without changes in maximal cardiac output [70]. Additionally, adaptations to VO_2_max following different training-intensity distribution (TID) interventions appear to depend on performance level, with competitive athletes responding more favorably to polarized TID and recreational athletes to pyramidal TID [71]. By contrast, the evidence for training-induced changes in SmO_2_min is less consistent. A recent meta-analysis [72] reported that endurance training did not significantly alter SmO_2_min values at the end of incremental tests, although this conclusion was limited by the small number of studies and heterogeneous protocols. Paquette et al. [73] conducted a study with competitive kayakers and found that HIIT produced performance improvements that were two to five times greater than those achieved with sprint interval training (SIT). Importantly, HIIT also induced greater peripheral adaptations, as reflected by lower muscle deoxygenation at submaximal intensity and lower SmO_2_min values. These findings suggest a potential overlap between the types of training that improve VO_2_max and those that affect SmO_2_min. However, particularly for SmO_2_min, future longitudinal studies are needed to validate its use as a practical marker of peripheral training adaptation, and well-controlled experimental interventions should be conducted to determine the most effective training sessions and TID for eliciting meaningful changes.

The negligible to weak correlations (r = −0.33 to 0.38) between kinematic parameters, such as CAD, SL, CT, and VO, and RE align with the conclusions of Barnes & Kilding [15], who stated that there does not appear to be a universally “efficient” movement pattern applicable to all runners, and with Pizzuto et al. [47], who reported non-significant relationships between RE and these spatiotemporal variables. Runners naturally acquire an optimal CAD and SL based on perceived exertion and repetitive exposure to specific running speeds [74,75]. Elite distance runners tend to exhibit lower VO and better RE compared to good runners [76,77]. This pattern aligns with Leite et al. [48], who reported that greater VO was associated with higher oxygen cost in recreational runners, consistent with the idea that lower VO accompanies better RE. However, Cavagna et al. [76] reported that excessively low VO can increase CAD and the internal work of contracting muscles, thereby decreasing RE. Taken together, these findings suggest that VO does not have a simple linear relationship with RE; rather, there may be an optimal range of VO that balances mechanical efficiency with metabolic cost. Lastly, with regard to CT, a trade-off between a midfoot strike and a longer contact time appears necessary, but this depends heavily on each runner’s physiology, training level, and biomechanical characteristics. For this reason, no clear correlation between CT and RE was identified [78]. Overall, these findings suggest that kinematic adjustments are not universally effective but rather highly individualized, and their potential impact on RE likely depends on each athlete’s specific characteristics.

A primary limitation of this study is the small and homogeneous sample size, which could limit the generalizability of the findings to broader athletic populations. Caution should be exercised before extrapolating these results to the individual disciplines that comprise the sport of triathlon. Previous research has suggested that VO_2_max values for running and cycling in national-level triathletes may be comparable to those observed in athletes specializing in these sports [79]. SmO_2_min in a specific muscle group appears to correlate with VO_2_max when the interrogated muscle substantially contributes to whole-body oxygen uptake, as indicated by differing correlations between vastus lateralis SmO_2_min and VO_2_max in cycling and running [33]. VO_2_max is widely regarded as a key determinant of whole-body endurance performance across sports [1], while SmO_2_min represents a relatively novel, muscle-specific marker that may reflect maximal oxygen extraction capacity at the local level [11]. An increasing number of studies have begun to explore the relationship between SmO_2_min and performance in various disciplines [34,35,36]. For example, Furno-Puglia et al. [35] reported that SmO_2_min was the strongest physiological predictor of mean power output during a time-trial handcycling test. Similarly, Paquette et al. [36] observed that SmO_2_min and delta deoxy[heme] together explained 90% of the variance in 200 m sprint performance among kayakers and canoeists. These findings imply that peripheral adaptations associated with reduced SmO_2_min values could be relevant to performance in multiple sports. Nonetheless, further investigations are needed, particularly in the individual sports that constitute triathlon, before firm conclusions can be drawn regarding the transferability of SmO_2_min as a performance marker across disciplines. Moreover, studies employing multiple NIRS probes would be valuable for identifying the muscle sites where SmO_2_min shows the strongest correlation with VO_2_max, depending on the specific sporting activity. Such insights could potentially establish SmO_2_min as a cost-effective indicator of cardiorespiratory fitness.

Another limitation of the study is that RE was calculated during a continuous incremental test following the recommendations of a preliminary investigation with a limited sample size [52]. It should be acknowledged that the progressive nature of this protocol may have introduced cumulative fatigue, potentially influencing RE measurements. Nevertheless, a preliminary study [52] suggested that when VO_2_ is averaged over the second half of each stage, the continuous incremental test can provide reasonable estimates of RE compared to a constant-load incremental protocol.

A further limitation is the small sample size (*n* = 11), which may lead to an overestimation of the observed R^2^. Although a post hoc power analysis indicated a power >0.99 for the large effect observed (R^2^ = 0.86, f^2^ = 6.14), these results should be interpreted with caution, as small samples can inflate effect sizes and limit the generalizability of the findings. However, the use of a homogeneous high-performance group minimized inter-individual variability and allowed us to examine physiological and kinematic determinants in a more controlled context.

## 5. Conclusions

The present study identified significant associations between physiological parameters and running performance. Vpeak was found to be largely explained (86%) by VO_2_max and SmO_2_min, suggesting that both variables are strongly associated with performance in trained triathletes. The results also support the potential utility of cost-effective NIRS technology for assessing SmO_2_min as a non-invasive marker associated with endurance capacity, maximal performance, and possibly muscle oxidative capacity. Additionally, the findings suggest that kinematic variables present high inter-individual variability and may not play a major role in explaining RE within this cohort.

## Figures and Tables

**Figure 1 sports-13-00316-f001:**
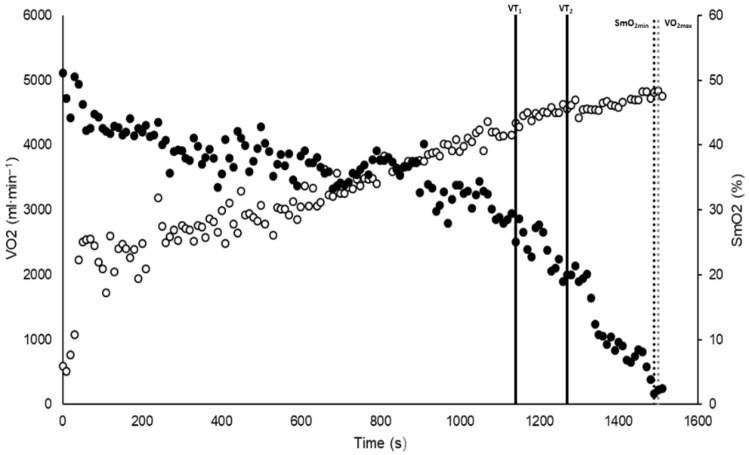
Representative time course of oxygen consumption (VO_2_) and muscle oxygen saturation (SmO_2_) during the VAM-EVAL test in a single subject. Open circles represent VO_2_ (mL·min^−1^, left y-axis), and closed circles represent SmO_2_ (%, right y-axis). Solid vertical black lines indicate the ventilatory thresholds VT1 and VT2, respectively. The dashed vertical black line marks the point of minimum SmO_2_ (SmO_2_min), while the dashed gray vertical line denotes maximal oxygen uptake (VO_2_max).

**Table 1 sports-13-00316-t001:** Descriptive values of all evaluated variables.

	Mean	SD	CV (%)	Range
Age (years)	24.3	5.3	22	19–33
Body weight (kg)	67.5	6.5	9.7	56–78
Body height (cm)	176.6	8.5	4.8	167–196
BMI (kg·m^−2^)	21.6	1.3	5.8	19.4–23.4
ATT (mm)	3.6	1.5	41.7	1.5–6.9
TTE (s)	1375.5	93.8	6.8	1250–1530
TTSmO_2_min (s)	1333.6	92.3	6.9	1230–1490
VO_2_max (mL·min^−1^·kg^−1^)	60.6	8.2	13.5	48–72.7
HRmax (bpm)	185.5	15.7	8.4	159–205
SmO_2_min (%)	15.1	13.3	88.4	0–38.6
VT1 (mL·min^−1^·kg^−1^)	51	6.5	12.8	39.6–64.8
VT2 (mL·min^−1^·kg^−1^)	57.9	7.8	13.5	45.4–69.9
RE12 (mL·min^−1^·kg^−1^)	42.3	3.5	8.4	34.7–46.1
CAD12 (spm)	164.6	5.3	3.2	156.3–177.1
VO12 (cm)	82.8	7	8.4	67.6–93.4
CT12 (ms)	243	7	2.9	232.1–252.6
SL12 (cm)	1255.8	38.9	3.1	1168.9–1304.4
RE16 (mL·min^−1^·kg^−1^)	55.1	6.4	11.7	44–64.4
CAD16 (spm)	174.1	5.6	3.2	164.6–185.4
VO216 (cm)	83.3	7.3	8.7	71.8–98.9
CT16 (ms)	199.3	5.6	2.8	191.9–206.9
SL16 (cm)	1568.2	59.9	3.8	1494.6–1706.6
Vpeak (km·h^−1^)	18.8	0.8	4	18–20

BMI, body mass index; ATT, adipose tissue thickness; TTE, time to exhaustion; TTSmO2min, time to SmO_2_min; VO_2_max, maximum oxygen consumption; HRmax, maximum heart rate; SmO_2_min, minimum muscle oxygen saturation; VT1, first ventilatory threshold; VT2, second ventilatory threshold; RE12, running economy at 12 km·h^−1^; CAD12, cadence at 12 km·h^−1^; VO12, vertical oscillation at 12 km·h^−1^; CT12, contact time at 12 km·h^−1^; SL12, stride length at 12 km·h^−1^; RE16, running economy at 16 km·h^−1^; CAD16, cadence at 16 km·h^−1^; VO16, vertical oscillation at 16 km·h^−1^; CT16, contact time at 16 km·h^−1^; SL16, stride length at 16 km·h^−1^; Vpeak, peak velocity.

**Table 2 sports-13-00316-t002:** Pearson correlation analysis between Vpeak and the analyzed physiological variables and RE and the analyzed kinematic variables.

	r	90% CI	*p*
Vpeak—VO_2_max	0.76	[0.38, 0.92]	0.007 *
Vpeak—SmO_2_min	−0.68	[−0.89, −0.25]	0.020 *
Vpeak—VT1	0.82	[0.51, 0.94]	0.002 *
Vpeak—VT2	0.70	[0.28, 0.90]	0.016 *
Vpeak—RE12	0.16	[−0.39, 0.63]	0.631
Vpeak—RE16	0.54	[0.03, 0.83]	0.083
Vpeak—HRmax	0.02	[−0.51, 0.54]	0.952
RE12—CAD12	0.24	[−0.33, 0.68]	0.483
RE12—VO12	−0.33	[−0.73, 0.23]	0.321
RE12—CT12	0.37	[−0.20, 0.75]	0.268
RE12—SL12	−0.19	[−0.65, 0.37]	0.573
RE16—CAD16	0.21	[−0.36, 0.66]	0.541
RE16—VO16	−0.26	[−0.69, 0.31]	0.445
RE16—CT16	0.38	[−0.18, 0.75]	0.250
RE16—SL16	−0.28	[−0.70, 0.29]	0.412
TTE—TTSmO_2_min	0.89	[0.70, 0.97]	<0.001 *
SmO_2_min—VO_2_max	−0.22	[−0.67, 0.35]	0.521

The level of significance (*p* < 0.05) between pairs of variables is marked with an asterisk (*). Vpeak, peak velocity; VO_2_max, maximum oxygen consumption; SmO_2_min, minimum muscle oxygen saturation; VT1, first ventilatory threshold; VT2, second ventilatory threshold; RE12, running economy at 12 km·h^−1^; RE16, running economy at 16 km·h^−1^; CAD12, cadence at 12 km·h^−1^; VO12, vertical oscillation at 12 km·h^−1^; CT12, contact time at 12 km·h^−1^; SL12, stride length at 12 km·h^−1^; CAD16, cadence at 16 km·h^−1^; VO16, vertical oscillation at 16 km·h^−1^; CT16, contact time at 16 km·h^−1^; SL16, stride length at 16 km·h^−1^; TTE, time to exhaustion; TTSmO_2_min, time to SmO_2_min.

**Table 3 sports-13-00316-t003:** Predictors of performance during maximal incremental VAM EVAL test (Vpeak).

Dependent Variable	R^2^	*p*	Indicator	β	*p*
Vpeak (km·h^−1^)	0.76	0.007	VO_2_max (mL·min^−1^·kg^−1^)	0.64	0.002
	−0.68	0.020	SmO_2_min (%)	−0.55	0.004

Vpeak, peak velocity; VO_2_max, maximum oxygen consumption; SmO_2_min, minimum muscle oxygen saturation.

## Data Availability

Our data are provided free of charge and can be accessed via the following link: https://doi.org/10.34810/data2352.

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
