# Peer review of "Physiological Predictors of Peak Velocity in the VAM-EVAL Incremental Test and the Role of Kinematic Variables in Running Economy in Triathletes"

_sports, 2025, doi:10.3390/sports13090316_

Round 1
Reviewer 1 Report
Comments and Suggestions for Authors
Dear Editors, thank you very much for the opportunity to have reviewed the manuscript “Physiological Predictors of Peak Velocity in the VAM-EVAL Incremental Test and the Role of Kinematic Variables in Running Economy in Triathletes” (Manuscript ID sports-3805798) for Sports journal (MDPI). The main objective of the study was to test the relationships between (i) physiological parameters and peak velocity; (ii) kinematic variables and running economy. Eleven participants were included. Main results indicated that peak velocity was influenced by VO₂max and muscle oxygen saturation while no correlations existed between kinematics and running economy. This work appears to have been well done by the authors. However, it has a low degree of originality (see for example: Leite, O. H. C., do Prado, D. M. L., Rabelo, N. D. D. A., Pires, L., Barton, G. J., Hespanhol, L., & Lucareli, P. R. G. (2024). Two sides of the same runner! The association between biomechanical and physiological markers of endurance performance in distance runners. Gait & posture, 113, 252–257. https://doi.org/10.1016/j.gaitpost.2024.06.027 ; Wiecha, S., Kasiak, P. S., Cieśliński, I., Maciejczyk, M., Mamcarz, A., & Śliż, D. (2022). Modeling Physiological Predictors of Running Velocity for Endurance Athletes. Journal of clinical medicine, 11(22), 6688. https://doi.org/10.3390/jcm11226688 ; Pizzuto, F., de Oliveira, C. F., Soares, T. S. A., Rago, V., Silva, G., & Oliveira, J. (2019). Relationship Between Running Economy and Kinematic Parameters in Long-Distance Runners. Journal of strength and conditioning research, 33(7), 1921–1928. https://doi.org/10.1519/JSC.0000000000003040 ). Because of this, I need to recommend against publication of the present manuscript.
Specific comments
P1L30. The introduction has been written with various standalone paragraphs. I recommend no more than 3 in order to improve the flow and interest for the potential readers.
P3L101. “The primary objective of the study was to evaluate the influence of physiological parameters—including maximal oxygen consumption, ventilatory thresholds, running economy, and minimum muscle oxygen saturation —on peak velocity. The secondary objective was to analyse the impact of kinematic variables—such as cadence, stride length, vertical oscillation, and contact time—on running economy.” – Lacking hypothesis
P3L117. “Eleven national-level triathletes voluntarily participated in the study.” – Missing sample size calculation
P9L343. “The findings of this study highlight important physiological and kinematic considerations for optimizing running performance.” – This conclusions seems not supported by the present experiment (cross-sectional).
Author Response
''Please see the attachment.''

Reviewer 2 Report
Comments and Suggestions for Authors
Writing Revision List
- Line 24. Should be written: "This study examined the influence of physiological parameters on peak velocity (Vpeak) and kinematic variables on running economy (RE) during an outdoor incremental VAM-EVAL test involving eleven national-level triathletes."
- Line 30. Should be written: "Links between RE and kinematic metrics were negligible to weak (r ≤ 0.38)."
- Line 36. Should be written: "VO₂max; SmO₂min; NIRS; Vpeak; incremental exercise testing."
- Line 52. Should be written: "RE is defined as the volume of oxygen required by an athlete to cover a given distance at a specific speed [1]."
- Line 62. Should be written: "Regarding RE, numerous studies have identified factors affecting this performance determinant, suggesting that RE reflects the interaction of physiological, biomechanical, anthropometric, and kinematic factors [14, 15]."
- Line 75. Should be written: "SmO₂ can be measured non-invasively during sports activity using continuous-wave near-infrared spectroscopy (NIRS) devices [21]."
- Line 85. Should be written: "The minimum muscle oxygen saturation (SmO₂min) has been observed to remain consistent within a session upon reaching exhaustion and can predict task failure during high-intensity efforts [28]."
- Line 100. Should be written: "Both velocities result from the interaction of physiological performance determinants specific to each athlete [1, 7]."
- Line 112. Should be written: "The secondary objective was to analyze the impact of kinematic variables—such as cadence, stride length, vertical oscillation, and contact time—on running economy."
- Line 135. Should be written: "The Moxy device was operated in default mode, cycling through four wavelengths 80 times every 2 seconds and averaging the readings to produce an output rate of 0.5 Hz."
- Line 151. Should be written: "The kinematic parameters—cadence, vertical oscillation, contact time, and stride length—were recorded throughout the incremental running test and averaged over 10-second intervals."
- Line 169. Should be written: "Participants were advised to consume a carbohydrate-rich diet in the 48 hours preceding the test."
- Line 191. Should be written: "Correlations were classified as negligible (0.00–0.30), weak (0.30–0.50), moderate (0.50–0.70), strong (0.70–0.90), or very strong (0.90–1.00) [48]."
- Line 224. Should be written: "The positive β coefficient for VO₂max indicates that better aerobic capacity is associated with higher Vpeak."
- Line 227. Should be written: "On the other hand, the kinematic parameters studied (cadence, vertical oscillation, contact time, and stride length) showed negligible to weak correlations with RE."
Scientific and Structural Review List
- L. 11-13. The authors should clarify whether the study’s findings generalize to other endurance athletes beyond triathletes, given the small, homogeneous sample.
- L. 28-30. The authors should justify why RE at 16 km·h⁻¹ was analyzed despite its moderate association (r = 0.54) with Vpeak, given the study’s focus on strong predictors.
- L. 42-44. The authors should provide a clearer distinction between VO₂max, RE, and physiological thresholds, as the current explanation overlaps conceptually.
- L. 71-73. The authors should cite recent studies validating NIRS-derived SmO₂ measurements in field settings, as current references (e.g., [22]) may not fully address technological advancements.
- L. 94-96. The authors should explicitly state whether Vpeak or MAS is more relevant for triathlon performance, given the discussion of their physiological differences.
- L. 124-126. The authors should justify the exclusion of athletes with adipose tissue thickness (ATT) > 7 mm, as this may limit applicability to broader populations.
- L. 146-148. The authors should clarify how averaging SmO₂min over a 10-second interval affects the precision of this metric, given its single-point use in analysis.
- L. 191-193. The authors should explain why stepwise regression was chosen over other methods (e.g., hierarchical regression) for analyzing predictors of Vpeak.
- L. 224-226. The authors should discuss whether the negative β coefficient for SmO₂min implies causation or merely association, given the observational design.
- L. 240-242. The authors should address potential confounding factors (e.g., muscle fiber composition) that may influence SmO₂min but were not measured.
- L. 273-275. The authors should reconcile the contradictory findings on vertical oscillation and RE, as some studies suggest lower oscillation improves economy.
- L. 298-300. The authors should explicitly state whether interventions targeting VO₂max and SmO₂min would differ from traditional endurance training.
- L. 310-312. The authors should acknowledge whether the incremental test protocol itself may have influenced RE measurements due to cumulative fatigue.
- L. 318-320. The authors should recommend future studies to validate SmO₂min as a practical marker for training adaptation, given the lack of longitudinal data.
- L. 325-327. The authors should clarify whether the negligible kinematic-RE correlations suggest that technique adjustments are irrelevant or simply individualized.
Author Response
''Please see the attachment''
All the comments in the writing revision list have been addressed according to the reviewer's recommendations. In the attached Word document, you will find our replies to each specific comment.

Round 2
Reviewer 1 Report
Comments and Suggestions for Authors
Thank you for the responses. However, based on the unanswered general comment that the manuscript seems to have a low degree of originality ("see for example: Leite, O. H. C., do Prado, D. M. L., Rabelo, N. D. D. A., Pires, L., Barton, G. J., Hespanhol, L., & Lucareli, P. R. G. (2024). Two sides of the same runner! The association between biomechanical and physiological markers of endurance performance in distance runners. Gait & posture, 113, 252–257. https://doi.org/10.1016/j.gaitpost.2024.06.027 ; Wiecha, S., Kasiak, P. S., Cieśliński, I., Maciejczyk, M., Mamcarz, A., & Śliż, D. (2022). Modeling Physiological Predictors of Running Velocity for Endurance Athletes. Journal of clinical medicine, 11(22), 6688. https://doi.org/10.3390/jcm11226688 ; Pizzuto, F., de Oliveira, C. F., Soares, T. S. A., Rago, V., Silva, G., & Oliveira, J. (2019). Relationship Between Running Economy and Kinematic Parameters in Long-Distance Runners. Journal of strength and conditioning research, 33(7), 1921–1928. https://doi.org/10.1519/JSC.0000000000003040 )."), it is my final decision to recommend against publication.
Author Response
''Please see the attachment''
